# COVID-19 Concerns and Personality of Commerce Workers: Its Influence on Burnout

Ana María Rodríguez-López and Susana Rubio-Valdehita *

Faculty of Psychology, Complutense University of Madrid, 20223 Madrid, Spain; anrodr17@ucm.es
* Correspondence: srubiova@ucm.es

**Abstract:** We analyze burnout in a sample of commercial workers in Spain and its relationship with sociodemographic variables, personality, and concern about the influence of the COVID-19 pandemic on their jobs through a cross-sectional design. Participants (*n* = 614) answered an online survey, including questions about sociodemographic data, concern, NEO-FFI (personality), and MBI (burnout syndrome). The survey took place from October 2020 to May 2021. We assessed the relationships between sociodemographic variables, pandemic concern, and personality as predictors of burnout by hierarchical regression analysis and then tested using SEM (structural equation models). The proposed model showed adequate goodness-of-fit indices. The results of the present study suggest that the COVID-19 pandemic had little effect to the development of burnout syndrome in commerce employees. However, in agreement with previous literature, the present study shows that personality has a significant role in predicting burnout. Neuroticism, introversion, conscientiousness, and agreeableness were strong predictors for burnout dimensions. In addition, we found that personality directly affected the pandemic concern: individuals with high levels of Neuroticism and low levels of extraversion, agreeableness, and conscientiousness have more pandemic concerns. In conclusion, personality is an important factor that affects the level of workers' concern about the influence of the pandemic on their job and the development of burnout syndrome. Furthermore, although we found significant differences between groups formed by various sociodemographic characteristics, the conclusion regarding this type of variable is that their ability to predict burnout is deficient.

**Keywords:** burnout; personality; COVID-19; pandemic; commerce sector; Spain





## 1. Introduction

COVID-19 emerged in December 2019 in Wuhan (Hubei, China), and on 11 March 2020, the WHO (World Health Organization) recognized it as a global pandemic [1]. In Spain, authorities identified the first local contagion on 26 February 2020, and since then, the emergence of the pandemic has seriously impacted the Spanish economy and labor market [2].

According to the Spanish Ministry of Industry, Commerce, and Tourism, the commerce sector constitutes one of the most valuable sectors of the Spanish economy [3]: it accounts for 12.6% of the total gross added value (GVA) of the Spanish economy, and retail trade alone accounts for 5.2% of the total GVA. Since the establishment of the state of alarm in March 2020, the restrictions adopted by the Spanish government and the governments of each autonomous community to fight COVID-19 have had a severe impact on all commercial activities. Consequently, the destruction of employment and an inevitable drop in the average income of consumers led to a contraction of domestic demand. Most commercial companies have faced a situation of reduced income for more than a year, which puts their permanence in the market at risk and has led to essential changes in the conditions of hundreds of thousands of jobs [4–6].

The delicate and complex situation faced by the commerce sector and its impact on its workers has led to tightening working conditions, translating into an increase in occupational diseases such as burnout [7–11]. Thus, the uncertainty caused by the

pandemic and the severe restrictions imposed to control the virus have led to dramatic changes in our daily lives and, consequently, in our jobs. Recent studies prove that these changes lead to stress, anxiety, burnout, fear, and frustration [12–14].

Some studies have analyzed how the COVID-19 pandemic impacts burnout levels in commerce workers, but they are few. For example, Rodríguez-López et al. [15] analyzed how the COVID-19 pandemic related to burnout levels in Spanish fashion retail workers, proving that fashion retail workers exhibited similar results as those observed I healthcare workers. The authors also observed that women's emotional burnout levels were higher than men's. It is necessary to emphasize that by carrying out their work in close contact with clients, commercial workers may face a situation similar to that of health workers in close contact with patients, which can also lead them to high levels of burnout [8,14,15]. Insomnia, depression, anxiety, uncertainty, and negative future expectations affected by the pandemic were substantial predictors for burnout [16,17].

Konstantopoulou et al. showed that commerce workers in Greece exhibited moderate to high burnout levels—higher in women and divorcees [18].

In South Korea, Hwang et al. analyzed burnout levels in a sample of workers from the commerce, banking, and hospitality sectors, and they found high levels of burnout in the three professional sectors. Younger and less experienced participants presented higher levels of burnout [19].

In the United States, Bakken and Winn analyzed burnout levels in a sample of 439 pharmacists and found that the COVID-19 pandemic increased burnout levels. [13]. Additionally, Toh et al. compared other essential workers (such as retail workers) to healthcare personnel and found that they exhibit higher depression, anxiety, and stress [14].

### 1.1. Personality and Burnout

Burnout is a syndrome that largely depends on working conditions. It is a psychosocial health disorder mainly associated with jobs that require direct contact with other people, such as clients, patients, or students. For this reason, it is more likely to appear in professional sectors, such as healthcare or education [17]. However, many studies have shown that specific personality characteristics are also closely related to burnout [20–25]. Most research on the relationships between personality and burnout has been developed under the Big Five model [26,27]. Robillard et al. showed that individuals with high scores in extraversion, conscientiousness, and neuroticism exhibited high levels of burnout [17]. Spagnoli et al. showed that neuroticism maintained a significant relationship with the development of burnout [20]. According to Robillard et al. and Venkatesh et al., high-conscientiousness individuals exhibited higher levels of burnout [17,21]. In general, the research has found that neurotic individuals, who are anxious, insecure, depressed, fearful, and nervous, are the most prone to burnout. They tend to exhibit greater emotional exhaustion due to their predisposition to negative feelings and their tendency to focus on the negative aspects of situations. Thus, when these feelings and perceptions are present at work, they tend to concentrate on the negative aspects of their performances and hold negative evaluations [9,10]. People with high scores in extraversion are more likely to experience positive emotions than introverts; thus, extroverts tend to positively view their job performance and build strong relationships at work, negatively related to burnout [11,23]. High agreeableness can also serve as a protective personality factor against burnout since very affable people tend to have positive views about their jobs and good interpersonal relationships at work [12,24]. Conscientious individuals have strong work ethics, are goal-oriented, and perceive themselves as productive [25], allowing them to feel insecure or anxious and, consequently, avoid burnout. Lastly, individuals with high scores in openness are open-minded and curious. This curiosity makes them perceive uncertain situations as exciting and attractive. On the other hand, close-minded individuals deal poorly with uncertainty, making them engage in unproductive behavior at work [24].

### 1.2. COVID-19 Concerns

In addition to Big-Five personality traits, several studies have discovered other variable clusters that predict burnout development [28,29] that may have also had a significant influence during the pandemic. For example, individuals with higher levels of risk perception engage more in protective behaviors [29,30] and therefore exhibit lower levels of COVID-19 concerns. However, people with a tendency to be active and sociable could present higher levels of COVID-19 concerns due to their difficulty in complying with mandatory social-distancing measures. Lee et al. investigated the relationship between mental health problems and unhealthy behaviors among healthcare workers in response to the COVID-19 pandemic [29]. They found that work-related stress and anxiety in response to the viral epidemic were associated with female sex, perception of the workplace as dangerous, and depressive symptoms.

Some studies have focused on studying the changes that COVID 19 has caused in the workplace and its effect on the well-being of workers and on the perceived risk of infection. The results of these studies show that companies have tried to adapt by implementing changes at the administrative level and also by providing personal protection measures against contagion [31]. These types of measures have reduced the perception of the risk of becoming infected [32] and have improved the well-being of employees [33].

### 1.3. Sociodemographic Variables

Moreover, several studies have found that some sociodemographic factors constituted significant predictors for burnout in healthcare and non-healthcare workers during the pandemic:

- Gender: women exhibit higher levels of risk perception and carry out more preventive actions [30].
- Marital status: divorcees exhibit higher levels of burnout [17].
- Age: younger workers exhibit higher levels of COVID-19 burnout [16].
- Children: having children constitutes a protective factor against burnout [34].
- Seniority: less experienced workers exhibit higher levels of COVID-19 burnout [35].
- Education level: individuals with higher education levels exhibit higher levels of COVID-19 burnout [36].
- Sick leaves: having a sick leave correlates with higher burnout levels [37,38].
- At work: changes in a work routine, lack of organizational justice, emotional work, interpersonal conflicts, environmental changes, performing uncommon tasks, and role conflicts. Various studies have shown that these variables are stressors that correlate with the development of burnout syndrome [39–42].

In conclusion, over the last two years, multiple studies have explored the development of COVID-19 burnout in specific labor sectors, such as the healthcare sector, and its relationship with the perception of the pandemic. However, few studies also include personality traits as possible predictors of the development of COVID-19 burnout. On the other hand, there are also few studies that sample workers in the commerce sector. Thus, the present study aims to assess the levels of burnout of commerce workers in the context of the pandemic in Spain. It aims to examine to what extent the worker's personality affects the levels of perceived burnout and find out whether concerns about how the pandemic may affect their job have any mediating effect on the relationship between personality and burnout. Our hypotheses were:

**Hypothesis 1 (H1).** *The participants will show moderated levels of burnout that will be highly related to their personality, so low scores in openness, agreeableness, extraversion, and conscientiousness and higher scores in neuroticism will correlate significantly with higher scores in burnout and pandemic concern.*

Recent studies on the development of burnout in workers other than healthcare workers (pharmacists, retail, commerce, banking, and hospitality workers) showed that partici-

pants exhibited moderate to high burnout levels during the COVID-19 pandemic [8,14–19]. Thus, we expect that our sample of Spanish commerce workers will exhibit similar burnout levels to their foreign peers. Furthermore, many studies showed that specific personality characteristics, such as neuroticism, agreeableness, extraversion and conscientiousness, are also closely related to burnout [20–25].

**Hypothesis 2 (H2).** *Higher scores in pandemic concerns will correlate significantly with higher burnout scores, but its effect will be less than that of personality.*

Several studies exhibited that the lack of perceived organizational support and the implementation of inappropriate measures to deal with the COVID-19 pandemic at work correlate with higher levels of anxiety, depression, and distress [32,33]. Furthermore, recent studies exhibited that anxiety, depression, and stress are consistent predictors for burnout [14,16,17]. Thus, we expect to find a positive correlation between pandemic concerns and higher levels of burnout in our sample of Spanish commerce workers. However, since the literature about the relationship between personality and burnout is more extensive and contrasted and as burnout is a syndrome that develops over time and therefore requires sustained maintenance of adverse working conditions over time [40], we expect that the association between burnout and pandemic concern will be less than that with personality.

**Hypothesis 3 (H3).** *Significant differences in the levels of burnout will be obtained according to certain sociodemographic features: age, gender, children, marital status, job contract, contact with the customer, seniority, level of education, and sick leaves in the last 12 months.*

Several studies have found that some sociodemographic factors (such as age, gender, children, marital status, job contract, contact with the customer, seniority, level of education, and sick leaves in the last year) constituted significant predictors for burnout in healthcare and non-healthcare workers during the pandemic [18,33–35]. Thus, we expect that our sample of Spanish commerce workers exhibit similar burnout levels to their foreign peers.

## 2. Materials and Methods

### 2.1. Participants and Procedure

We established a cross-sectional design. We recruited a sample of 614 participants through LinkedIn, Instagram, and Twitter, all working in the commerce sector (convenience sampling). To participate, it was necessary to have a job seniority of at least six months. Thus, participants accessed a link that led to a Google Forms questionnaire. Then, they gave their informed consent to participate. We collected data from October 2020 to May 2021. The mean age of the participants was 32.82 years old (*SD* = 8.60), ranging from 18 to 65. Table 1 shows the sociodemographic characteristics of the participants. We classified the different jobs occupied by the participants into two groups, according to whether they implied a direct relationship with customers (shop assistants, cashiers, head sales assistant, store boss) or not (managers, human resource technicians, marketing, visual merchandising, accounting).

### 2.2. Instruments

Data were collected using the following instruments:

Sociodemographic questionnaire: this questionnaire included questions about age, gender, educational level, civil status, whether they have children or not, employment position, sick leaves during the last year, job seniority, and type of contract.

**Table 1.** Characteristics of the sample.

|  |  | **N** | **%** |
|---|---|---|---|
| Gender | Woman | 400 | 65.1 |
|  | Man | 214 | 34.9 |
| Type of job | In contact with customers | 322 | 52.4 |
|  | Not in contact with customers | 292 | 47.6 |
| Education | University | 418 | 68.1 |
|  | Secondary | 196 | 31.9 |
| Marital Status | Single/divorced | 356 | 58.0 |
|  | Married/stable partner | 258 | 42.0 |
| Have children | No | 455 | 74.1 |
|  | Yes | 159 | 25.9 |
| Job Seniority | Between 6 months and 1 year | 152 | 24.8 |
|  | Between 1 and 2 years | 119 | 19.4 |
|  | Between 2 and 5 years | 177 | 28.8 |
|  | More than 5 years | 166 | 27.0 |
| Type of contract | Temporal | 153 | 24.9 |
|  | Permanent | 461 | 75.1 |
| Work on weekends | No | 137 | 22.3 |
|  | Yes | 477 | 77.7 |
| Sick leave | No | 446 | 72.6 |
|  | Yes | 168 | 27.4 |

MBI: We used the Spanish version of the Maslach Burnout Inventory to measure burnout syndrome [43,44]. The questionnaire contains the following subscales: (1) emotional exhaustion (experiences of being emotionally exhausted by the demands of work), (2) depersonalization (exhibition of cold attitudes and detachment in the workplace), (3) personal accomplishment (presence of feelings of self-efficiency and accomplishment in the workplace). MBI meets the criteria for factor validity and internal consistency [43,44]. In our sample, the Cronbach's $\alpha$ for MBI showed good reliability both for the total scale ($\alpha = 0.80$) and for one of the burnout dimensions ($\alpha = 0.89$ for emotional exhaustion; $\alpha = 0.72$ for depersonalization; and $\alpha = 0.77$ for personal accomplishment). Items of the MBI appear in Appendix A.

Concerns about COVID-19 questionnaire: Participants answered 3 questions about how they perceived the pandemic could affect their job. Each item has a Likert response scale from "totally disagree" (0) to "totally agree" (5). We describe the items in Table 2. This questionnaire's factor analysis (main components) showed a unifactorial structure that explained 72.78% of the variance. The Cronbach's $\alpha$ was 0.81, showing outstanding reliability.

**Table 2.** Mean and standard deviation (*SD*) for all measures.

|  |  | **Mean** | *SD* |
|---|---|---|---|
| COVID-19 (range: 1–5) | Item 1. The COVID-19 pandemic creates uncertainty in my job | 3.80 | 1.43 |
|  | Item 2. I fear that my work situation will be affected by the pandemic | 3.99 | 1.34 |
|  | Item 3. I think my work situation is going to get worse due to the COVID-19 pandemic | 3.70 | 1.36 |
| Personality (range: 1–100) | Neuroticism | 61.33 | 35.51 |
|  | Extroversion | 49.58 | 35.61 |
|  | Openness | 53.14 | 34.71 |
|  | Conscientiousness | 43.54 | 34.69 |
|  | Agreeableness | 39.30 | 32.64 |
| Burnout (range: 1–100) | Emotional exhaustion | 62.07 | 29.26 |
|  | Depersonalization | 52.46 | 29.95 |
|  | Personal accomplishment | 48.18 | 27.42 |

NEO-FFI: The Spanish version of NEO-FFI was used to evaluate the five major personality traits: neuroticism, extraversion, openness, friendliness, and responsibility. NEO-FFI meets the criteria for factor validity and internal consistency [45]. For our sample, we obtained adequate reliability: $\alpha$(N) = 0.85; $\alpha$(E) = 0.87; $\alpha$(O) = 0.83; $\alpha$(A) = 0.75; $\alpha$(C) = 0.82. Items of the NEO-FFI appear in Appendix B.

The ethics committee of the authors' research center approved the study (Ref.: 2019/20-022).

### 2.3. Data Analysis

To explore the influence of sociodemographic variables, concerns about the pandemic, and personality on burnout, we computed three hierarchical regression analyses for each burnout dimension (emotional exhaustion, depersonalization, and personal accomplishment). Hierarchical linear regression is a particular form of multiple linear regression analysis in which more variables are added to the model in separate steps called blocks. We use it to statistically control certain variables and see whether adding variables significantly improves a model's ability to predict the criterion variable. In the first step, we introduced the sociodemographic variables; in the second step, we added personality traits; and finally, we included the responses to the items of pandemic concerns.

Later, we tested the resulting model using SEM (structural equation models) to explore the structural relations between the variables and the possible mediating effect of job-related pandemic concern. We used the maximum likelihood estimation method and the Mardia coefficient considering multivariate normality when its critical ratio is equal to or less than than 1.96 [46]. We checked the goodness-of-fit indices, such as the comparative fit index (CFI), root mean square error of approximation (RMSEA), and standardized root mean residual (SRMR) [47,48]. In addition, we studied the magnitude of $\chi2$ divided by its degrees of freedom (CMIN/DF) that must be less than 3 for a reasonable adjustment. CFI assesses the extent to which the tested model repeats the observed covariance matrix [46]. A cutoff criterion of CFI > 0.90 is needed to ensure that mis-specified models are not accepted [49]. The RMSEA introduces a correction for lack of parsimony: we accepted a cutoff value close to 0.08 [50] for an appropriate fit. The SRMR is an index of the averaged standardized residuals between the observed and hypothesized covariance matrices [51]. We interpreted SRMR values less than 0.08 as a good fit [52]. We also considered other indices: corrected goodness index (AGFI), goodness of fit (GFI), and normed fit index (NFI). The values of these indices should be close to 0.90 or above to be considered a good fit [52]. There were no missing cases in the sample, so using any imputation method was unnecessary.

Finally, to delve into the study of the possible effect of some sociodemographic variables on burnout and facilitate comparison with the results obtained in previous research using this methodology, *t*-tests were computed.

We performed all statistical analyses using IBM SPSS Statistical Package 25.0 and AMOS 22.0.

## 3. Results

Table 2 shows the mean and standard deviation for all measures. Mean scores for the perception of the impact of the pandemic revealed that participants are concerned about its possible effects on their work.

Scores indicating high burnout levels are 50 or more for emotional exhaustion and depersonalization and 50 or less for personal accomplishment, so participants exhibit moderate levels of burnout. Even though our sample does not exhibit high levels of burnout, it nearly does. Regarding personality, participants exhibited average scores for the five personality domains.

Table 3 shows the results of the hierarchical regression analysis. The results indicate that the personality variables (especially neuroticism) were the ones that showed the most significant relationship with burnout. Neuroticism, extraversion, and agreeableness were significant predictors for burnout. More neurotic and less extroverted and pleasant

participants scored higher in burnout. Openness and conscientiousness were not related to burnout.

**Table 3.** Hierarchical regression analysis results.

| Step | | Emotional Exhaustion | | | | Depersonalization | | | | Personal Accomplishment | | | |
|---|---|---|---|---|---|---|---|---|---|---|---|---|---|
| | | $\beta$ | $R^2$ | $\Delta R^2$ | $\Delta F$ | $\beta$ | $R^2$ | $\Delta R^2$ | $\Delta F$ | $\beta$ | $R^2$ | $\Delta R^2$ | $\Delta F$ |
| | | | 0.06 | 0.06 | 3.85 ** | | 0.07 | 0.07 | 4.84 ** | | 0.12 | 0.12 | 8.21 ** |
| | Job position | −0.06 | | | | −0.08 | | | | 0.28 ** | | | |
| | Age | −0.02 | | | | −0.10 | | | | 0.05 | | | |
| | Gender | −0.07 | | | | −0.10 | | | | 0.03 | | | |
| | Education | −0.03 | | | | 0.04 | | | | 0.01 | | | |
| 1 | Marital status | 0.06 | | | | 0.01 | | | | 0.04 | | | |
| | Children | −0.09 | | | | −0.07 | | | | 0.09 | | | |
| | Antiqueness | 0.10 | | | | 0.10 | | | | −0.04 | | | |
| | Contract | −0.06 | | | | −0.06 | | | | 0.04 | | | |
| | Work on weekend | 0.06 | | | | 0.05 | | | | 0.18 ** | | | |
| | Sick leave | 0.15 ** | | | | 0.06 | | | | −0.05 | | | |
| | | | 0.22 | 0.16 | 25.13 ** | | 0.21 | 0.13 | 20.06 ** | | 0.30 | 0.18 | 30.11 ** |
| | Neuroticism | 0.28 ** | | | | 0.25 ** | | | | −0.24 ** | | | |
| | Extraversion | −0.13 * | | | | −0.07 | | | | 0.23 ** | | | |
| 2 | Openness | 0.07 | | | | 0.05 | | | | 0.01 | | | |
| | Agreeableness | −0.12 ** | | | | −0.19 ** | | | | 0.11 * | | | |
| | Conscientiousness | −0.10 | | | | −0.05 | | | | 0.08 | | | |
| | | | 0.25 | 0.03 | 7.60 ** | | 0.23 | 0.02 | 6.29 ** | | 0.30 | 0.00 | 0.75 |
| | The COVID-19 pandemic creates uncertainty in my job | 0.09 | | | | 0.06 | | | | 0.02 | | | |
| 3 | I fear that my work situation will be affected by the pandemic | −0.14 * | | | | −0.16 * | | | | 0.05 | | | |
| | I think my work situation is going to get worse due to the pandemic | 0.18 ** | | | | 0.19 ** | | | | −0.06 | | | |

\* $p < 0.05$; \*\* $p < 0.001$.

The effect of the sociodemographic variables was somewhat higher concerning personal accomplishment but lower concerning the other two burnout domains. We found a significant correlation between having had a sick leave in the last year and emotional exhaustion, so participants who had a sick leave exhibited higher levels of emotional exhaustion. Additionally, participants with an administrative job (managers, human resource technicians), not in direct contact with customers and not working on the weekend exhibited significantly higher levels of personal accomplishment.

Job-related pandemic concern had some influence, although minimal, on emotional exhaustion and depersonalization and no influence on personal accomplishment.

The results obtained through hierarchical regression analysis elaborate a model to test the relationships between personality, pandemic concern, and burnout. The model appears in Figure 1. We did not add sociodemographic variables to the model, as they did not obtain a significant weight in the prediction of burnout.

The critical ratio for the Mardia coefficient was 1.40, showing the multivariate normality of the data. This model (Figure 1) showed adequate goodness-of-fit indices: CMIN/DF = 2.39, RMSEA = 0.048 (LO90 = 0.034, HI90 = 0.062); CFI = 0.974; SRMR =0.0545; AGFI = 0.954, GFI = 0.978; NFI = 0.956. In this model, all the regression weights are significant except for openness and personality ($r = 0.58$, $p = 0.386$) and between concern and burnout ($r = 0.43$, $p = 0.460$). The results of the SEM analysis are generally consistent with those obtained by hierarchical regression. They show that personality has significant direct effects on burnout ($r = 0.884$, $p < 0.001$), while the direct effects of pandemic concern are deficient and not significant. Neuroticism was the personality dimension with the greatest predictive power, while openness was not related to burnout. Additionally, personality had a direct effect on job-related pandemic concern ($r = 0.178$, $p < 0.001$). More neuroticism and less extraversion, agreeableness, and conscientiousness correlated with more burnout and more pandemic concern. The indirect effects of pandemic concern on burnout, mediated by personality, were low and nonsignificant.

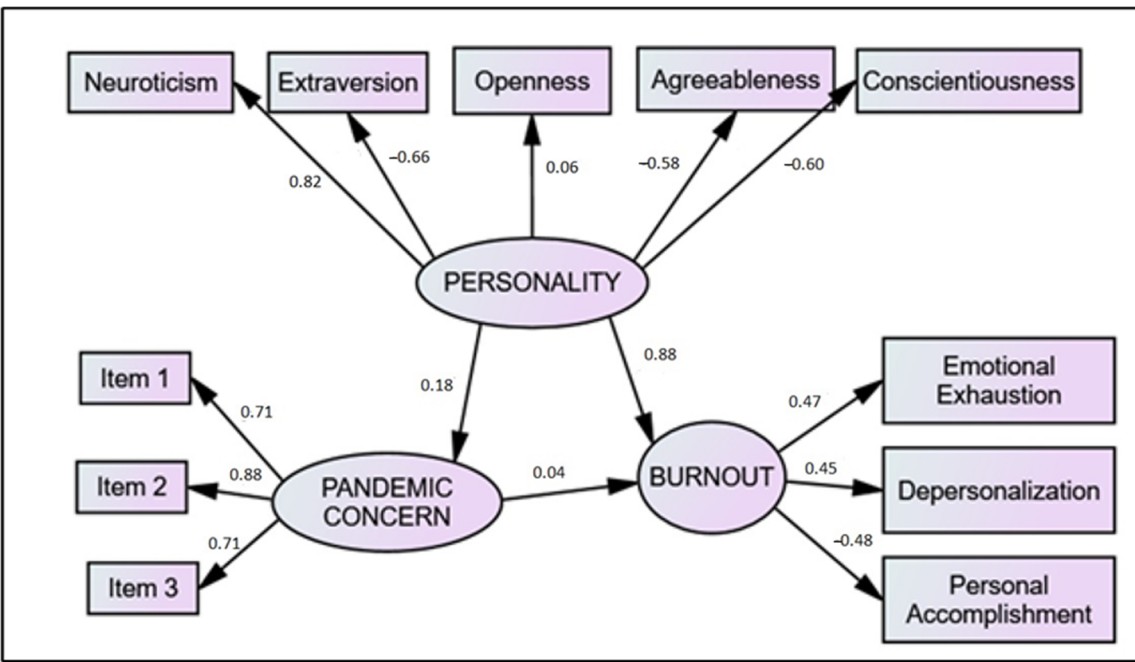

**Figure 1.** Standardized regression weights from SEM. Item 1: The COVID-19 pandemic creates uncertainty in my job; Item 2: I fear that my work situation will be affected by the pandemic; Item 3: I think my work situation is going to get worse due to the COVID-19 pandemic.

*T*-tests were computed to examine the differences in the burnout dimension between groups established from sociodemographic variables. Mean and standard deviations are presented in Table 4. Significant differences between type of job were obtained for emotional exhaustion ($t = 2.92$, $p = 0.004$), depersonalization ($t = 4.47$, $p < 0.001$) and personal accomplishment ($t = -6.99$, $p < 0.001$), showing that participants who have a job in direct contact with customers present more burnout. Men and women obtained significant differences in emotional exhaustion ($t = 2.43$, $p = 0.015$) and depersonalization ($t = 3.84$, $p < 0.001$) but no difference in personal accomplishment ($t = -1.38$, $p = 0.169$), showing women experience more burnout. No significant differences were found for education ($p > 0.05$ in all cases). Regarding marital status, significant differences were found in depersonalization ($t = 2.13$, $p = 0.033$) and personal accomplishment ($t = -3.52$, $p < 0.001$) but no difference in emotional exhaustion ($t = 0.65$, $p = 0.518$), showing single/divorced people experience more burnout. Significant differences were found between those participants who have children and those who do not for emotional exhaustion ($t = 2.38$, $p = 0.018$), depersonalization ($t = 3.23$, $p = 0.001$), and personal accomplishment ($t = -4.26$, $p < 0.001$), showing less burnout experienced by those who have children. Participants who have a permanent employment contract showed less burnout, resulting from statistically significant differences in depersonalization ($t = 2.22$, $p = 0.027$) and personal accomplishment ($t = -3.21$, $p = 0.001$) but not in emotional exhaustion ($t = 0.99$, $p = 0.320$). Differences between working on weekends and not were obtained in emotional exhaustion ($t = -2.62$, $p = 0.009$) and depersonalization ($t = -2.96$, $p = 0.003$) but not in personal accomplishment ($t = -1.36$, $p = 0.175$). Participants who must work on weekends showed higher scores in the three dimensions of burnout. Finally, participants who have had a sick leave in the last year exhibited more burnout, significantly in emotional exhaustion ($t = -4.28$, $p < 0.001$) and depersonalization ($t = -2.66$, $p = 0.008$) but not in personal accomplishment ($t = 1.55$, $p = 0.122$).

**Table 4.** Mean and SD for burnout dimensions in groups established from sociodemographic variables.

| | | Emotional Exhaustion | | Depersonalization | | Personal Accomplishment | |
|---|---|---|---|---|---|---|---|
| | | Mean | SD | Mean | SD | Mean | SD |
| Job | In contact with customers | 65.33 | 30.58 | 57.52 | 29.03 | 41.09 | 26.92 |
| | Not in contact with customers | 58.46 | 27.32 | 46.86 | 30.00 | 56.00 | 25.84 |
| Gender | Woman | 64.15 | 31.00 | 55.81 | 29.64 | 47.07 | 27.62 |
| | Man | 58.15 | 25.30 | 46.18 | 29.57 | 50.26 | 26.99 |
| Education | University | 62.44 | 28.58 | 51.68 | 30.19 | 47.85 | 26.77 |
| | Secondary | 61.25 | 30.71 | 54.10 | 29.44 | 48.88 | 28.81 |
| Marital status | Single/divorced | 62.71 | 29.84 | 54.64 | 29.78 | 44.90 | 27.47 |
| | Married/stable partner | 61.16 | 28.47 | 49.43 | 29.98 | 52.71 | 26.75 |
| Children | No | 63.71 | 28.97 | 54.74 | 29.44 | 45.43 | 27.09 |
| | Yes | 57.33 | 29.64 | 45.89 | 30.52 | 56.05 | 26.93 |
| Contract | Temporal | 64.10 | 31.02 | 57.09 | 28.79 | 42.05 | 28.87 |
| | Permanent | 61.38 | 28.65 | 50.91 | 30.20 | 50.21 | 26.64 |
| Weekends | No | 56.31 | 27.59 | 45.83 | 30.29 | 45.37 | 27.56 |
| | Yes | 63.71 | 29.54 | 54.35 | 29.61 | 48.98 | 27.36 |
| Sick leave | No | 59.00 | 29.55 | 50.49 | 29.97 | 49.23 | 28.11 |
| | Yes | 70.19 | 26.90 | 57.66 | 29.35 | 45.39 | 25.38 |

## 4. Discussion and Conclusions

The present study suggests that the COVID-19 pandemic affects the development of burnout syndrome in commerce employees but in a very minimal and irrelevant way. However, according to previous literature, the present study shows that personality plays a significant role in the development burnout. Thus, individuals who tend to be neurotic, hostile, and introverted are prone to developing burnout. People who exhibit these personality traits are inclined to focus on the adverse aspects of their performance, detrimentally hold negative evaluations, deal poorly with uncertainty, and tend to engage in unproductive behavior at work [17,20,21,23–25]. In agreement with previous research, neuroticism seemed to be the strongest predictor for emotional exhaustion and depersonalization [17,20]. Consistent with the meta-analysis by Alarcón and Swider on the relationship between personality and burnout, extraversion was also the strongest predictor for personal accomplishment, and openness exhibited the weakest association with burnout [17,23,24]. Furthermore, we found that personality directly affected pandemic concern; individuals with high levels of neuroticism and low scores in extraversion, agreeableness, and conscientiousness have more pandemic concerns.

Regarding sociodemographic variables and previous literature, individuals who had a sick leave in the last year exhibited higher levels of emotional exhaustion. Thus, number of sick leaves in the past 12 months correlate with higher emotional exhaustion and depersonalization levels. Recent studies found that burnout increases the risk of future absences because of mental, physical, and behavioral disorders [53–56].

In contrast to findings obtained in previous studies in the last two years, we found no significant relationships between age, gender, marital status, children, and job seniority with burnout. This allows us to conclude that these sociodemographic variables are not weighted enough to reliably predict burnout levels. The apparent discrepancy with results from other research may be due to the different methodology used to analyze the data. In our study, we explored the strength of the relationships between sociodemographic variables and the dimensions of burnout. In contrast, in previous studies, the questions have been directed toward comparing groups with ANOVA or *t*-tests. The outlook is different in both cases. Suppose we use the contrast of means between the generated groups according to the sociodemographic variables to analyze data. In that case, we also obtained significant results for all variables considered, except for education, in line with the findings of previous studies. Women presented higher levels of burnout than men [30], married participants exhibited lower levels of burnout than single or divorced participants,



younger and less experienced employees displayed higher levels of burnout than older employees [16,35], having children represented a protective factor against burnout [34], and finally, having taken a sick leave was related to higher burnout levels [38,39]. The conclusion that we can obtain from our study regarding the influence of sociodemographic variables on burnout is that even with significant differences between the groups, these types of variables are not capable of predicting, in an acceptable way, burnout that a worker can experience. Moreover, this conclusion is more evident if we consider that in regression analysis, the sociodemographic variables were only able to explain 6% of emotional exhaustion, 7% of depersonalization, and 12% of personal accomplishment.

Our study provides information on how the COVID-19 pandemic impacts workers in the commercial sector, one of the most valuable sectors in the Spanish economy, expanding scientific knowledge beyond the effects of the pandemic on healthcare personnel. The results obtained in this study allow us to conclude that job-related pandemic concern influences the variables of burnout of emotional exhaustion and depersonalization but very minimally. When we introduced the responses to the items about concern in the regression analysis, the increase in $R^2$ only concerned the introduction of the personality. However, the statistically significant predictive power increased 3% for emotional exhaustion and 2% for depersonalization, with no improvement in the predictive power for personal accomplishment. The results of the SEM corroborated this small influence. In this sense, it is necessary to consider several aspects. First, although employees in the commercial sector are concerned about the effects that the pandemic may have on their future employment and their working conditions have been affected by the economic crisis associated with the restrictive measures imposed by governments to fight against the pandemic, its impact has been objectively less than that suffered by healthcare workers. In addition, we must bear in mind that burnout is a syndrome that develops over time and therefore requires sustained maintenance of adverse working conditions over time [43]. Therefore, our data cannot possibly reveal the increase in burnout that the pandemic can cause in commercial employees. Therefore, even though this influence appears to be minimal in this study, we consider that it is important to continue to pay attention to burnout in commercial workers in order to avoid its increase in the future, should the economic effects of pandemic persist, and even more so if we consider that our sample displayed moderate levels of burnout.

Finally, the results of this study show the critical role that the employee's personality plays in the development of burnout syndrome. In agreement with previous research, neuroticism, extraversion, conscientiousness, and agreeableness, were shown to be good predictors of burnout, while we obtained no relationships between openness and the dimensions of burnout [25,53]. Despite this, it is essential to highlight that according to our results, personality explains 16% of emotional exhaustion, 13% of depersonalization, and 18% of personal accomplishment, which allows us to conclude that burnout also depends on other factors: work conditions. Therefore, in future studies, it is necessary to investigate the structure of the relationships that exist between the employee's personality traits and working conditions, both as predictors of burnout. In this sense, during the pandemic, many workers have endured working conditions that can increase the appearance of burnout, such as fear of being infected by others in the work setting, lack of protective equipment, reduction in the workforce, decrease in social support, increased workload, etc.

The results of our study have valuable practical implications. Although conditions in the work environment certainly impact burnout, our findings suggest that burnout is also associated with worker personality. Therefore, even when organizations use burnout interventions focused on changing the work environment by reducing or eradicating job stressors, some people may still experience high levels of burnout due to their personalities. Organizations might use personality assessment to identify personnel who are prone to burnout and use this information to determine which employees would likely benefit most from stress-reduction training or to identify which employees should or should not be given stressful jobs.

This study has made it possible to expand our knowledge about how personality and other sociodemographic characteristics of employees in the Spanish commercial sector affect the development of burnout in times of the COVID-19 pandemic. However, the study has some limitations. Although the sample used is made up of 614 individuals, we consider that it should be extended. Thus, a future line of research could be to test our results in other countries. Furthermore, a future line of research could be to address different personality models to study the development of COVID-19 burnout. Moreover, 65% of the sample has a university degree. This is not representative of the Spanish population (41% of the Spanish population has a university degree). In addition, the sampling method used, convenience sampling, prevents the generalization of the results obtained to the entire Spanish population of workers in the commerce sector. Thus, a future line of research should ensure that the sample is representative of the population of all sociodemographic factors evaluated and correct the biases that may cause deviations. Finally, our study shows a fixed image of the current situation, so it would be beneficial to carry out longitudinal studies that could give a better impression of whether the effects of the pandemic change over time.

**Author Contributions:** Conceptualization, A.M.R.-L. and S.R.-V.; methodology, A.M.R.-L. and S.R.-V.; validation A.M.R.-L. and S.R.-V.; formal analysis, A.M.R.-L.; investigation, A.M.R.-L.; resources, S.R.-V.; data curation, A.M.R.-L. and S.R.-V.; writing—original draft preparation, A.M.R.-L.; writing—review and editing, A.M.R.-L. and S.R.-V.; visualization, A.M.R.-L. and S.R.-V.; supervision, S.R.-V.; project administration, S.R.-V. All authors have read and agreed to the published version of the manuscript.

**Funding:** This research received no external funding.

**Institutional Review Board Statement:** The study was conducted according to the guidelines of the Declaration of Helsinki and approved by Ethics Committee of Faculty of Psychology of Complutense University of Madrid (protocol code 2019/20-022, September 2020).

**Informed Consent Statement:** Informed consent was obtained from all subjects involved in the study.

**Data Availability Statement:** The data presented in this study are available on request from the corresponding author.

**Acknowledgments:** We are grateful to the participants for taking part in this study.

**Conflicts of Interest:** The authors declare no conflict of interest.

## Appendix A. Items of MBI: Maslach Burnout Inventory

Indicate how frequently the following statements apply to you and add the points indicated on top of the respective box:

0. Never
1. At least a few times a year
2. At least once a month
3. Several times a month
4. Once a week
5. Several times a week
6. Every day

1. I feel emotionally exhausted because of my work
2. I feel worn out at the end of a working day
3. I feel tired as soon as I get up in the morning and see a new working day stretched out in front of me
4. I can easily understand the actions of my colleagues/supervisors
5. I get the feeling that I treat some clients/colleagues impersonally, as if they were objects
6. Working with people the whole day is stressful for me
7. I deal with other people's problems successfully
8. I feel burned out because of my work

9.　I feel that I influence other people positively through my work
10.　I have become more callous to people since I have started doing this job
11.　I'm afraid that my work makes me emotionally harder
12.　I feel full of energy
13.　I feel frustrated by my work
14.　I get the feeling that I work too hard
15.　I'm not really interested in what is going on with many of my colleagues
16.　Being in direct contact with people at work is too stressful
17.　I find it easy to build a relaxed atmosphere in my working environment
18.　I feel stimulated when I been working closely with my colleagues
19.　I have achieved many rewarding objectives in my work
20.　I feel as if I'm at my wits' end
21.　In my work I am very relaxed when dealing with emotional problems
22.　I have the feeling that my colleagues blame me for some of their problems

**Appendix B. Items of NEO-FFI**

The next questions are about you and the way you are. Answer with sincerity marking your degree of agreement according to the following scale:
1. Strongly disagree
2. Disagree
3. Neutral
4. Agree
5. Strongly agree

1.　I often feel inferior to others.
2.　I am a cheerful and spirited person.
3.　Sometimes when I read poetry or look at a work of art, I feel deep emotion or excitement.
4.　I tend to think the best of people.
5.　I never seem to be able to organize myself.
6.　I rarely feel scared or anxious.
7.　I really enjoy talking to people.
8.　Poetry has little or no effect on me.
9.　Sometimes I bully or flatter people into doing what I want.
10.　I have clear objectives and I strive to achieve them in an organized manner.
11.　Sometimes scary thoughts come to mind.
12.　I enjoy parties where there are a lot of people.
13.　I have a wide variety of intellectual interests.
14.　Sometimes I trick people to do what I want.
15.　I work hard to achieve my goals.
16.　Sometimes it seems to me that I am worth absolutely nothing.
17.　I don't consider myself particularly cheerful.
18.　I am curious about the forms I find in art and in nature.
19.　If someone starts to fight with me, I am also willing to fight.
20.　I have a lot of self-discipline.
21.　Sometimes things seem too bleak and hopeless.
22.　I like having a lot of people around.
23.　I find philosophical discussions boring
24.　When I have been offended, what I try is to forgive and forget.
25.　Before taking an action, I always consider its consequences.
26.　When I'm under heavy stress, sometimes I feel like I'm going to break down.
27.　I am not as lively or as encouraging as other people.
28.　I have a lot of fantasy.
29.　My first reaction is to trust people.
30.　I try to do my homework carefully so that it doesn't have to be done again.
31.　I often feel tense and restless.

32.   I am a very active person.
33.   I like to focus on a dream or fantasy and letting it grow and develop, explore all its possibilities.
34.   Some people think of me that I am cold and calculating.
35.   I strive to achieve perfection in everything I do.
36.   At times I have felt bitter and resentful.
37.   In meetings, I generally prefer others to speak.
38.   I have little interest in thinking about the nature of the universe or the human condition.
39.   I have a lot of faith in human nature.
40.   I am efficient and effective in my work.
41.   I am quite stable emotionally.
42.   I run away from the crowds.
43.   Sometimes I lose interest when people talk about very abstract issues.
44.   I try to be humble.
45.   I am a productive person who always finishes his work.
46.   I am seldom sad or depressed.
47.   Sometimes I ooze happiness.
48.   I experience a great variety of emotions or feelings.
49.   I believe that most of the people I deal with are honest and trustworthy.
50.   Sometimes I act first and then think.
51.   Sometimes I do things impulsively and then I regret it.
52.   I like to be where the action is.
53.   I often try new foods or foods from other countries.
54.   I can be sarcastic and scathing if necessary.
55.   There are so many little things to do that sometimes what I do is not attend to any of them.
56.   It's hard for me to lose my temper.
57.   I don't really like chatting with people.
58.   I rarely experience strong emotions.
59.   Beggars do not inspire me sympathy.
60.   Many times, I do not repair in advance what I have to do.

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
