# Peer review of "COVID-19 Concerns and Personality of Commerce Workers: Its Influence on Burnout"

_sustainability, doi:10.3390/su132212908_

Round 1

Reviewer 1 Report

1. The impact of COVID-19 pandemic on workers/employees' wellbeing at the workplace can be further explained as the background context, addressing the existing policies at workplace to manage workers' wellbeing issues. The authors shall add a few literature to consolidate this part:

Wong, E.L.Y., Ho, K.F., Wong, S.Y.S., Cheung, A.W.L., Yau, P.S.Y., Dong, D. and Yeoh, E.K., 2020. Views on workplace policies and its impact on health-related quality of life during coronavirus disease (COVID-19) pandemic: cross-sectional survey of employees. International journal of health policy and management.

Hou, H.C., Remøy, H., Jylhä, T. and Putte, H.V., 2021. A study on office workplace modification during the COVID-19 pandemic in The Netherlands. Journal of Corporate Real Estate.

Mihalache, M. and Mihalache, O.R., 2021. How workplace support for the COVID‐19 pandemic and personality traits affect changes in employees' affective commitment to the organization and job‐related well‐being. Human Resource Management.

2. The authors didn't elaborate the measurement of "burnout" and "personalities". I suggest the authors list the original survey questions in the manuscript or attach the questionnaire as an appendix to indicate the measurement items. Otherwise the validity of the measurement is questionable. 

3. The hypotheses are not well presented. Please refer to relevant paper to present the hypotheses in a proper way. Also, the development of the six hypotheses lack detailed literature support. Do you develop these hypotheses because they were not investigated before? or the relationships among the the constructs are more significant in the context of COVID-19? what are the novel part of your hypotheses development and the theoretical model you proposed?

4. if you have developed six hypotheses, why not illustrate a proposed theoretical model? The model was presented as a results, while a proposed form can be integrated with the hypotheses development. 

5. what are the difference between pandemic concern and burnout? please list the questions. I wonder how the respondents can distinguish these two concepts? 

6. as for the data collection, the sample size appears to be too small to reflect the whole population who work in the commerce sector in Spain. Please address why you adopt this data collection approach. 

Author Response

First of all, we would like to warmly thank the reviewer for his comments and the interest and effort made in reviewing our manuscript.

  1. The impact of COVID-19 pandemic on workers/employees' wellbeing at the workplace can be further explained as the background context, addressing the existing policies at workplace to manage workers' wellbeing issues. The authors shall add a few literature to consolidate this part:

We have included the references provided by the reviewer in the manuscript.

  1. The authors didn't elaborate the measurement of "burnout" and "personalities". I suggest the authors list the original survey questions in the manuscript or attach the questionnaire as an appendix to indicate the measurement items. Otherwise the validity of the measurement is questionable. 

As reviewer recommend, original survey questions for MBI and NEO-FFI have been included in Appendix A and Appendix B.

  1. The hypotheses are not well presented. Please refer to relevant paper to present the hypotheses in a proper way. Also, the development of the six hypotheses lack detailed literature support. Do you develop these hypotheses because they were not investigated before? or the relationships among the the constructs are more significant in the context of COVID-19? what are the novel part of your hypotheses development and the theoretical model you proposed?
  2. if you have developed six hypotheses, why not illustrate a proposed theoretical model? The model was presented as a results, while a proposed form can be integrated with the hypotheses development. 

Following suggestions 3 and 4 of the reviewer, the section dedicated to hypotheses has been modified to better fit the proposed model. We have reduced the hypotheses to three and have added a text justifying them below each one.

  1. what are the difference between pandemic concern and burnout? please list the questions. I wonder how the respondents can distinguish these two concepts? 

While the pandemic concern refers to the fear that workers have about how the pandemic may affect their job, and was evaluated through a three-item questionnaire, burnout is a work syndrome that occurs when the workers perceive that they have been subjected to adverse working conditions for some time that generate emotional fatigue, depersonalization, and low personal fulfillment. Burnout was assessed using Maslach's MBI. All survey items have been included in an appendix.

  1. as for the data collection, the sample size appears to be too small to reflect the whole population who work in the commerce sector in Spain. Please address why you adopt this data collection approach. 

We have included in the Participants section a mention of the type of sampling used, and also, by using non-probability sampling we have included this aspect as a limitation of our study at the end of the manuscript.

Reviewer 2 Report

Dear authors,

Thank you for the edits based on the reviewer's feedback. I believe you have done an incredible amount of work. You have applied most of my comments statisfactory.

I do have one major concern. Multiple t-tests have been performed, without any correction for multiple testing. Performing all these t-test in an exploratory study can lead to the occurrence of false positives. My advice would be to describe the results of table 4 descriptively, without testing, and focus on the exploratory nature of the study.

A few minor comments:

  1. Can you add a footnote to figure 1 explaning the meaning for item 1, 2 and 3?
  2. Can you do a layout check of Table 3? 

Author Response

Thank you for the edits based on the reviewer's feedback. I believe you have done an incredible amount of work. You have applied most of my comments statisfactory.

I do have one major concern. Multiple t-tests have been performed, without any correction for multiple testing. Performing all these t-test in an exploratory study can lead to the occurrence of false positives. My advice would be to describe the results of table 4 descriptively, without testing, and focus on the exploratory nature of the study.

First of all, we would like to warmly thank the reviewer for his comments and the interest and effort made in reviewing our manuscript.

We agree with the suggestion made by the reviewer, but we consider that it is better to present the results of the comparisons since our purpose with this analysis is to compare with the results of other previous investigations in which these comparisons tests are performed.

A few minor comments:

  1. Can you add a footnote to figure 1 explaning the meaning for item 1, 2 and 3?

          We have added the footnote in the figure, as indicated by the reviewer.

     2. Can you do a layout check of Table 3? 

        Table 3 has been checked.